# Inter-regional disparities in emergency department utilization among critically ill patients: A nationwide study from South Korea

**Mi Ra Oh[1], Young Jin Huh[1], Han Na Lee[1], Se Hyung Kim[1], Sung Min Lee[2]\***

1 National Emergency Medical Center of National Medical Center, Seoul, Korea, 2 Department of Emergency Medicine, Chonnam National University Medical School and Chonnam National University Hospital, Gwangju, Korea

\* terran034@gmail.com

## Abstract

### Background

This study examined inter-regional emergency department (ED) utilization among critically ill patients in South Korea and identified demographic, socioeconomic, and system-level factors.

### Methods

A retrospective analysis of 741,701 critically ill ED visits in 2021 was conducted using the National Emergency Department Information System (NEDIS). Inter-regional utilization was defined as receiving care outside the residential emergency medical service (EMS) region. Patient characteristics were compared between inter- and intra-regional groups. Mixed-effects logistic regression, modified Poisson regression, and average marginal effects (AMEs) were applied to identify associated factors. The results were presented concurrently using odds ratios (ORs), adjusted risk ratios (aRRs), and AMEs, including 95% confidence intervals (CIs).

### Results

Overall, 21.7% of critically ill patients received inter-regional care. Male patients had higher odds of inter-regional utilization (OR 1.08, 95% CI: 1.07–1.10; aRRs 1.05, 95% CI: 1.03–1.08), although their absolute probability was significantly lower (AME −8.29%p, 95% CI: −12.42 to −4.16), reflecting differences between relative and absolute measures. Adults aged 19–44 years (OR 1.35, 95% CI: 1.32–1.39) and 45–64 years (OR 1.24, 95% CI: 1.21–1.27) showed higher odds of inter-regional utilization, although corresponding AMEs were not statistically significant. Among older adults, the 65–74-year group showed a non-significant decrease in absolute probability (AME −0.90%p, 95% CI: −3.39 to 1.58), whereas the 75–84-year group demonstrated

**Data availability statement:** The NEDIS data used in this study are not publicly available due to privacy restrictions imposed by the National Emergency Medical Center. Access may be requested through the National Medical Center (https://www.nmc.or.kr) subject to ethical approval.

**Funding:** This research was supported by a grant of National Medical Center, Republic of Korea (grant number: NMC2023-PR-06). There was no additional external funding received for this study.

**Competing interests:** The authors have declared that no competing interests exist.

a significant reduction (AME –5.78%p, 95% CI: –7.64 to –3.92). Pediatric patients also exhibited elevated odds of inter-regional use. Medical Aid beneficiaries had lower relative odds (OR 0.65, 95% CI: 0.63–0.67), and AMEs showed no statistically significant absolute differences. Inter-hospital transfer (OR 1.20, 95% CI: 1.16–1.24) was associated with higher inter-regional use, with a statistically significant increase in absolute probability (AME + 3.74%p, 95% CI: 0.86–6.63). Arrival by ambulance or private vehicle showed positive but non-significant AMEs. High-acuity patients (KTAS 1–2) and those transferred as their final disposition also demonstrated higher relative odds of inter-regional utilization.

## Conclusions

Inter-regional ED utilization was more likely among men, younger and middle-aged adults, high-acuity patients, and those arriving by ambulance or transferred from another hospital, while Medical Aid patients were less likely to receive such care. These findings highlight the importance of strengthening local emergency capacity, optimizing referral and transfer pathways, and implementing region-specific strategies alongside nationwide initiatives.

## Introduction

Critically ill emergency patients face a high risk of mortality and require immediate definitive care at medical institutions equipped to perform advanced procedures and surgical interventions. To meet this need, the Ministry of Health and Welfare of South Korea designated 29 emergency medical service (EMS) regions based on residential areas and established an appropriate number of regional emergency medical centers for each region to ensure efficient resource allocation [1]. However, hospitals capable of delivering initial treatment for severe emergency conditions are predominantly concentrated in urban areas, limiting timely access to care for residents in suburban or rural regions [2].

This geographic imbalance is not unique to Korea. As similar disparities exist globally. A recent Finnish study reported that the use of inter-hospital transfers by prehospital EMS units placed a significant burden on EMS readiness, particularly in rural areas with limited medical infrastructure [3]. Similarly, the U.S. Centers for Disease Control and Prevention reported disparities in EMS access between urban and rural areas, with longer response times and reduced access to advanced life support in rural regions negatively impacting clinical outcomes [4]. Such disparities contribute to the inter-regional utilization of emergency departments (ED) by critically ill patients. Although the timely provision of EMS can significantly reduce mortality and disability rates, delays in receiving definitive care due to inter-regional transfers may lead to poorer clinical outcomes [5]. Moreover, the uneven distribution of emergency medical resources can hinder effective service delivery. Ensuring equitable and efficient allocation of resources across EMS regions is essential to improve continuity of care and patient outcomes [6].

In response to this, the Ministry of Health and Welfare introduced a policy in 2021 aimed at establishing a "regionally integrated emergency care system," that aimed to provide treatment for emergency patients within their own EMS region [7]. Despite this initiative, there has been a lack of empirical studies examining whether critically ill patients are indeed receiving care within their designated region. Previous studies have demonstrated regional disparities in EMS utilization and patient transfer patterns in Korea, classifying service areas into inflow-, mixed-, and outflow-types based on patient movement indices [8–10]. Internationally, the network equity-quality model has been proposed to conceptualize hospital transfer networks and their implications for care equity [11]. In addition, healthcare utilization has often been analyzed using Andersen's behavioral model, which highlights the roles of predisposing, enabling, and need factors in shaping access to care [12].

Therefore, this study aims to analyze the inter-regional utilization patterns of critically ill emergency patients and identify factors influencing such movements. By clarifying these determinants, we seek to provide evidence that can inform both region-specific strategies and nationwide policies, ultimately ensuring more equitable and efficient EMS planning.

## Methods

### Data source

This study analyzed 741,701 cases of ED visits for critical emergency conditions from 01/01/2021–31/12/2021, using data from the National Emergency Department Information System (NEDIS)—a nationwide registry operated by the National Emergency Medical Center under the National Medical Center. The data were accessed for research purposes on 05/01/2023. It collects real-time clinical data from emergency medical institutions across the country. All data are anonymized at the point of collection, ensuring that no personally identifiable information was included. Patients were classified as critically ill if their diagnosis matched one of the 1,120 severe disease codes defined across 28 disease categories by the National Emergency Medical Center in Table 1. These conditions, which may be medical, surgical, or trauma-related in nature, are characterized by a substantial risk of mortality and require timely interventions during the acute phase due to their significant impact on patient outcomes. As of 2021, Korea's emergency medical system comprised 29 EMS regions, including 39 regional emergency medical centers and 127 local emergency medical centers in Table 2.

### Study setting

Variables analyzed in this study included residential address, address of the treating emergency medical institution, and general ED visit characteristics such as sex, age, type of emergency medical institution, insurance type, disease status, route of visit, mode of arrival, final triage classification, clinical outcome, and ED length of stay (LOS), and hospital LOS. Based on EMS region data, cases were classified as "matched (0)" if the patient's residential EMS region corresponded to the EMS region of the treating hospital, and "unmatched (1)" if they differed.

Sex was categorized as male or female, and age was grouped as 0–4 years, 5–12 years, 13–18 years, 19–44 years, 45–64 years, 65–74 years, 75–84 years, and ≥85 years to capture more granular age-related variation in inter-regional emergency care utilization. Emergency medical institutions were classified as regional or local centers. Insurance type was categorized as National Health Insurance, Medical aid, or other. Disease status was categorized as "disease" or "non-disease." Route of visit included direct visit, transfer from another facility, or other. The 'other' category included visit routes that did not fit into the predefined classifications, such as atypical or unclassified cases. Mode of arrival was categorized as ambulance, private vehicle, or other.

Triage level was assessed using the Korean Triage and Acuity Scale (KTAS), categorized as KTAS 1–2, KTAS 3, and KTAS 4–5. Final clinical outcomes were categorized as discharge, transfer, death, or other. ED LOS was measured in minutes from arrival to ED discharge, and hospital LOS was measured in days from admission to discharge.

**Table 1. Classification of critical illness by Disease group and diagnostic code (KCD-8).**

| No. | KCD-8 Codes | Disease Category |
|---|---|---|
| 1 | I210-I219 | Acute myocardial infarction |
| 2 | I6300-I64 | Acute stroke |
| 3 | I610-I629 | Intracranial hemorrhage |
| 4 | I600-I609 | Subarachnoid hemorrhage |
| 5 | ICISS_2015 <= 0.90 | Major trauma (based on severity index) |
| 6 | I7101-I7109, I7110-I7119, I713, I715, I718 | Aortic dissection |
| 7 | K8000-K8011, K8030-K8041, K8051, K810, K819, K830, K831 | Hepatobiliary diseases |
| 8 | K352-K353, K631, K650-K659, K661 | Surgical diseases |
| 9 | I8500-I8501, I864, I983, K920-K922, K226, K2500, K2540, K2501, K2521, K2541, K2561, K260, K262, K264, K266, T181 | Gastrointestinal bleeding/foreign bodies |
| 10 | R042, R048, R049, T1740-T1799 | Tracheobronchial bleeding/foreign bodies |
| 11 | T360-T659 | Intoxication (including CO poisoning) |
| 12 | O000-O009, O140-O159, O4200, O4201, O4209, O4210, O4211, O4219, O4220, O4221, O4229, O4290, O4291, O4299, O450-O459, O6000-O6039, O800-O809, O820-O829, O720-O723, O622 | Peripartum diseases |
| 13 | P0700-P0739, P220-P229, P240-P249, P360-P369, P520-P529, P590-P599 | Preterm/low birth weight |
| 14 | T3130-T3199, T2030-T2039, T2070-T2079, T213, T217 | Major burn |
| 15 | G410-G419 | Status epilepticus |
| 16 | A830-A879, G000-G07 | Meningitis |
| 17 | A021, A227, A241, A267, A400-A409, A410-A414, A419, A427, B007, B377 | Sepsis |
| 18 | E1000-E1018, E1100-E1118, E1300-E1318, E1400-E1418 | Diabetic coma |
| 19 | I260, I269, I802 | PTE/DVT |
| 20 | I441, I442, I450-I459, I472, I480-I489, I490, I495, I498, I499 | Arrhythmia |
| 21 | J80, J81, J850-J869, J9600-J9699 | ARDS/pneumonia/pulmonary edema |
| 22 | D65 | DIC |
| 23 | K561-K563, K565-K566 | Intussusception/intestinal obstruction |
| 24 | S480-S489, S580-S589, S6800-S689, S780-S789, S880-S889, S980-S984, T050-T059, T060-T068, T116, T136 | Amputation |
| 25 | N170-N179 | Acute kidney injury |
| 26 | H3300-H3309, H3310-H332, H3330-H334, H3350-H3358, H340-H349, H400, H4010-H4019, H4020-H403, H404, H405, H406, H4080-H409, H420, H428 | Ophthalmic emergencies |
| 27 | I460-I469 | Post resuscitation status |
| 28 | N44, N4500-N4502, N4590-N4592 | Urological emergencies |

KCD: Korean Standard Classification of Diseases; CO: carbon monoxide; PTE: pulmonary thromboembolism; DVT: deep vein thrombosis; ARDS: acute respiratory distress syndrome; DIC: disseminated intravascular coagulation.

## Analytical methods

This study compared patterns of in-region and inter-regional EMS utilization among critically ill patients. For EMS regions with high rates of inter-regional utilization, network analysis was used to visualize the geographic distribution and volume of inter-regional patient movement, highlighting areas with concentrated patient flows between specific regions. Network analysis was conducted to illustrate inter-regional patient flows, with nodes representing EMS regions and edges weighted by the proportion of transferred patients.

Differences in general characteristics by EMS region movement status were analyzed using the chi-squared test, with results presented as frequencies and percentages. Because ED and hospital LOS were not normally distributed, the

**Table 2. The 29 Emergency Medical Service regions in Korea.**

| No. | EMS region | Constituent areas |
|---|---|---|
| 1 | Seoul Northwest | Jongno-gu, Jung-gu, Yongsan-gu, Eunpyeong-gu, Mapo-gu, Seodaemun-gu |
| 2 | Seoul Northeast | Nowon-gu, Dongdaemun-gu, Jungnang-gu, Seongbuk-gu, Gangbuk-gu, Dobong-gu, Namyangju (Gyeonggi) |
| 3 | Seoul Southwest | Yangcheon-gu, Gangseo-gu, Guro-gu, Geumcheon-gu, Yeongdeungpo-gu, Dongjak-gu, Gwanak-gu, Gwangmyeong (Gyeonggi) |
| 4 | Seoul Southeast | Songpa-gu, Seongdong-gu, Gwangjin-gu, Seocho-gu, Gangnam-gu, Gangdong-gu, Guri, Hanam, Yangpyeong (Gyeonggi) |
| 5 | Busan | Busan, Gimhae, Yangsan, Miryang, Geoje (Gyeongnam) |
| 6 | Daegu | Daegu, Gyeongsan, Goryeong, Gunwi, Seongju, Yeongcheon, Cheongdo (Gyeongbuk), Geochang, Hapcheon (Gyeongnam) |
| 7 | Incheon | Incheon (except Ganghwa), Bucheon, Siheung |
| 8 | Gwangju | Gwangju, Gangjin, Gokseong, Naju, Damyang, Boseong, Yeonggwang, Jangseong, Jangheung, Hampyeong, Hwasun (Jeonnam), Gochang, Sunchang (Jeonbuk) |
| 9 | Daejeon | Daejeon, Sejong, Gyeryong, Gongju, Geumsan, Nonsan, Buyeo, Cheongyang (Chungnam), Yeongdong, Okcheon (Chungbuk), Muju (Jeonbuk) |
| 10 | Ulsan | Ulsan |
| 11 | Gyeonggi Northwest | Goyang, Gimpo, Paju (Gyeonggi), Ganghwa (Incheon) |
| 12 | Gyeonggi Northeast | Uijeongbu, Dongducheon, Yangju, Yeoncheon, Pocheon (Gyeonggi), Cheorwon (Gangwon) |
| 13 | Gyeonggi Southwest | Suwon, Ansan, Osan, Hwaseong, Anyang, Gwacheon, Gunpo, Uiwang (Gyeonggi) |
| 14 | Gyeonggi Southeast | Seongnam, Gwangju, Yongin, Icheon (Gyeonggi) |
| 15 | Gangwon East | Gangneung, Goseong, Donghae, Samcheok, Sokcho, Yangyang, Jeongseon, Taebaek, Pyeongchang (Gangwon) |
| 16 | Gangwon Chuncheon | Chuncheon, Yanggu, Inje, Hongcheon, Hwacheon (Gangwon), Gapyeong (Gyeonggi) |
| 17 | Wonju-Chungju | Wonju, Yeongwol, Hoengseong (Gangwon), Yeoju (Gyeonggi), Chungju, Danyang, Jecheon (Chungbuk) |
| 18 | Chungnam Cheonan | Cheonan, Dangjin, Seosan, Asan, Yesan, Taean, Hongseong (Chungnam), Anseong, Pyeongtaek (Gyeonggi) |
| 19 | Chungbuk Cheongju | Cheongju, Goesan, Boeun, Eumseong, Jeungpyeong, Jincheon (Chungbuk) |
| 20 | Jeonbuk Iksan | Iksan, Gunsan (Jeonbuk), Seocheon, Boryeong (Chungnam) |
| 21 | Jeonbuk Jeonju | Jeonju, Gimje, Namwon, Buan, Wanju, Imsil, Jangsu, Jeongeup, Jinan (Jeonbuk) |
| 22 | Jeonnam Mokpo | Mokpo, Muan, Shinan, Yeongam, Wando, Jindo, Haenam (Jeonnam) |
| 23 | Jeonnam Suncheon | Suncheon, Goheung, Gwangyang, Gurye, Yeosu (Jeonnam) |
| 24 | Gyeongbuk Andong | Andong, Mungyeong, Bonghwa, Yeongyang, Yeongju, Yecheon, Uiseong, Cheongsong (Gyeongbuk) |
| 25 | Gyeongbuk Gumi | Gumi, Gimcheon, Chilgok, Sangju (Gyeongbuk) |
| 26 | Gyeongbuk Pohang | Pohang, Gyeongju, Yeongdeok, Uljin, Ulleung (Gyeongbuk) |
| 27 | Gyeongnam Changwon | Changwon, Uiryeong, Changnyeong, Haman (Gyeongnam) |
| 28 | Gyeongnam Jinju | Jinju, Goseong, Namhae, Sacheon, Sancheong, Tongyeong, Hadong, Hamyang (Gyeongnam) |
| 29 | Jeju | Jeju, Seogwipo |

EMS: Emergency medical service.

non-parametric Mann–Whitney U test was used for comparisons, with results expressed as median, the first quartile (Q1), and the third quartile (Q3).

To identify factors influencing inter-regional utilization, multivariable logistic regression analysis was performed. Because patients were clustered within the same EMS regions, we applied a mixed-effects logistic regression model with EMS region included as a random intercept to account for the clustering structure. For sensitivity analysis, we additionally fitted a standard logistic regression model and calculated cluster-robust standard errors at the EMS region level. Furthermore, given that the prevalence of inter-regional utilization in our study was relatively high

(approximately 22%), and odds ratios (ORs) may overestimate the effect size, we performed modified Poisson regression to estimate adjusted risk ratios (aRRs). The results are now presented in parallel as ORs, average marginal effects (AMEs), and aRRs, including 95% confidence intervals (CIs). Some AMEs did not reach statistical significance and should be interpreted cautiously as complementary measures. To evaluate model validity, multicollinearity was assessed using variance inflation factors (VIFs).

Additionally, subgroup logistic regression analyses were performed for selected EMS regions with high inter-regional utilization to identify region-specific characteristics influencing inter-regional ED use and to provide evidence to support the development of targeted policy interventions. All statistical analyses were performed using IBM SPSS Statistics, version 27 (IBM Corp., Armonk, NY, USA) and R version 4.2.3 (https://www.r-project.org/). A $P$-value of <0.05 denoted statistical significance.

### Ethics statement

This study was approved for exemption from ethical review by the Institutional Review Board of the National Medical Center (NMC-2023-01-004). This study is a database analysis of anonymized patient visits; thus, no patient consent was required. This study adhered to the STROBE reporting guideline [13].

## Results

### Inter-regional utilization trends of critically ill emergency patients

We analyzed regional utilization patterns of critically ill emergency patients by comparing the location of their ED visits with their registered residential EMS regions. The findings revealed notable regional disparities in the proportion of patients receiving care within their own EMS regions. The highest ~~out-of-region~~ inter-regional ED utilization rates were observed in the following six regions: Gyeonggi Northeast (42.6%), Seoul Northeast (39.9%), Gyeongbuk Gumi (36.5%), Jeonnam Mokpo (34.8%), Seoul Northwest (34.6%), and Jeonnam Suncheon (33.8%). In each of these regions, over 30% of critically ill emergency patients receive care outside of their home EMS region, highlighting substantial inter-regional patient flow (Fig 1).

We further examined the inter-regional movement of patients in the six EMS regions with within-region utilization rates below 70%. In Gyeonggi Northeast, 16.8% of patients traveled to Seoul Northeast for emergency care, 19.8% of patients residing in Seoul Northeast sought treatment in Seoul Southeast, and 24.3% of patients from Gyeongbuk Gumi moved to Daegu. In the Jeonnam region, 24.8% of patients from Mokpo and 20.8% from Suncheon utilized EDs in Gwangju, while 11.7% of patients from Seoul Northwest were treated in Seoul Southwest (Fig 2).

### Characteristics of the study population

Of the 741,701 critically ill emergency patients, 160,700 (21.7%) received care outside their residential EMS region (Table 3). Inter-regional utilization was more common among men (57.1% vs. 42.9%). Age-specific analysis revealed a bimodal pattern: higher proportions among younger (0–4 and 5–12 years) and middle-aged adults (19–44 and 45–64 years), whereas patients aged ≥85 years were least likely to receive care outside their EMS region ($p < 0.001$). Patients with National Health Insurance were more likely to use inter-regional services, whereas Medical Aid beneficiaries were less likely ($p < 0.001$). Inter-regional patients were more often transferred from another hospital (31.1% vs. 21.3%) and less often direct visits (64.1% vs. 75.0%; $p < 0.001$). They also included a higher proportion of high-acuity cases (KTAS 1–2: 27.6% vs. 24.3%) and more often had hospital transfer as their final disposition (15.0% vs. 12.9%). Notably, inter-regional patients had significantly longer stays both in the ED (median 305 vs. 258 minutes) and in hospital (7.7 vs. 7.0 days; $p < 0.001$).

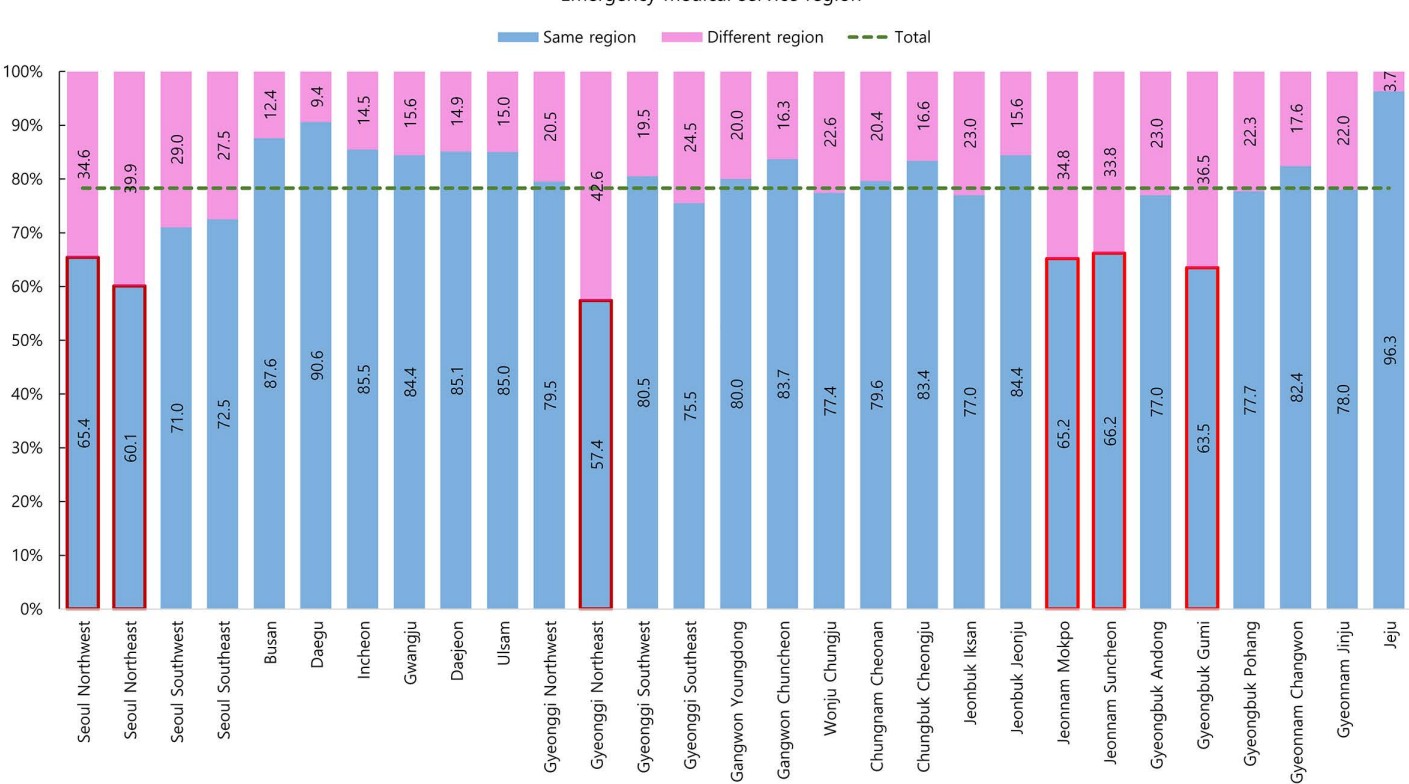

**Fig 1. Proportion of critically ill patients utilizing emergency departments within their designated emergency medical service regions.** Regions with less than 70% within-region utilization are highlighted, indicating high rates of inter-regional transfer.

### Factors associated with inter-regional utilization in critically ill emergency patients

We examined factors influencing inter-regional utilization among critically ill emergency patients using ORs, aRRs, and AMEs (Table 4). Male patients exhibited higher odds of inter-regional utilization compared with females (OR 1.08, 95% CI: 1.08–1.10; aRRs 1.07, 95% CI: 1.04–1.09). However, the AME indicated that the absolute probability of inter-regional use was on average 8.3 percentage points lower among men than women, highlighting a divergence between relative and absolute measures. This discrepancy underscores the importance of interpreting AMEs alongside ORs and aRRs to capture both relative and absolute perspectives. Age was also associated with inter-regional movement. Adults aged 19–44 years (OR 1.35, 95% CI: 1.32–1.39) and 45–64 years (OR 1.24, 95% CI: 1.21–1.27) had higher odds than those aged ≥85 years, although their AMEs were not statistically significant. Among older adults, the 65–74-year group showed a non-significant decrease in absolute probability (AME –0.90%p, 95% CI: –3.39 to 1.58), whereas the 75–84-year group demonstrated a significant reduction (AME –5.78%p, 95% CI: –7.64 to –3.92). Pediatric patients also exhibited elevated odds of inter-regional use. Insurance type showed a strong association. Medical Aid beneficiaries had a significantly lower likelihood of inter-regional utilization across all effect measures (OR 0.64, 95% CI: 0.62–0.67; aRR 0.74, 95% CI: 0.67–0.82; AME –5.90%p, 95% CI: –7.73 to –3.99). National Health Insurance beneficiaries showed a nonsignificant positive AME.

Patients with disease-related conditions had a higher absolute probability of inter-regional use (AME + 1.40%p, 95% CI: 0.84–1.95). Arrival-related factors also played an important role. Inter-hospital transfer was associated with higher relative odds (OR 1.20, 95% CI: 1.16–1.24) and a statistically significant increase in absolute probability (AME + 3.74%p, 95% CI: 0.86–6.63). Direct visits showed an even larger absolute probability difference (+5.39%p).

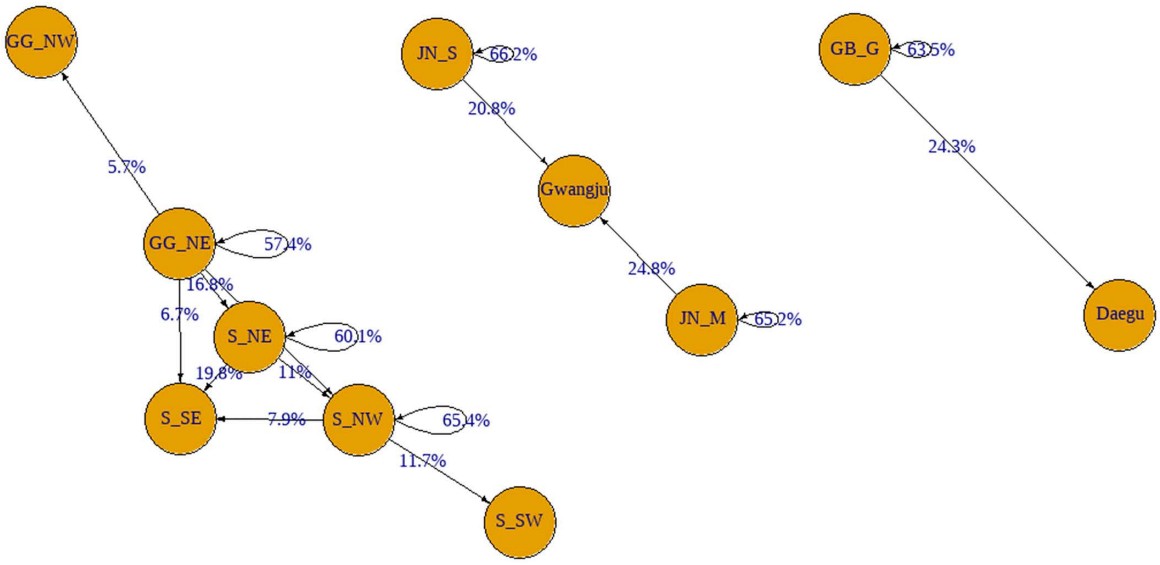

**Fig 2. Distribution of inter-regional transfers of critically ill emergency patients (EDs) from emergency medical service (EMS) regions with high rates of inter-regional ED utilization.** Nodes represent EMS regions, while the arrows indicate the direction and proportion (%) of patient transfers. S_NW: Seoul Northwest, S_SW: Seoul Southwest. S_SE: Seoul Southeast, S_NE: Seoul Northeast, GG_NW: Gyeonggi Northwest, GG_NE: Gyeonggi Northeast, JN_M: Jeonnam Mokpo, JN_S: Jeonnam Suncheon, GB_G: Gyeongbuk Gumi.

Regarding transportation, ambulance (AME + 2.91%p, 95% CI: –0.73 to 6.54) and private vehicle arrivals (AME + 0.78%p, 95% CI: –0.87 to 2.42) displayed positive but non-significant AMEs. High-acuity patients (KTAS 1–2) had higher relative odds (OR 1.21, 95% CI: 1.18–1.23), although corresponding AMEs were non-significant. Patients whose final disposition was transfer also showed elevated odds (OR 1.25, 95% CI: 1.17–1.33), with a small, non-significant increase in absolute probability. All VIF values were below 5, indicating no multicollinearity.

### Subgroup analysis of factors contributing to high inter-regional

**ED utilization.** We conducted additional multivariable logistic regression analyses for the three high-inflow regions identified in Fig 2 (i.e., Seoul Southeast, Gwangju, and Daegu) to explore the characteristics of patients who utilized emergency services outside their designated EMS regions. As shown in the Table 5, the factors associated with inter-regional utilization differed across regions. In Seoul–Southeast, higher odds were observed among male patients, a broad range of middle- and older-aged adults (45–74 years), and patients with higher acuity (KTAS 1–3). In Gwangju, pediatric patients demonstrated notably high odds across all age groups under 18 years, and visits to regional emergency medical centers (REMCs) showed the strongest association with inter-regional utilization. Patients transferred from another hospital also had significantly elevated odds. In Daegu, the highest odds were observed among younger adults (19–44 years), with elevated risks also present among pediatric groups. Additionally, both ambulance and private vehicle arrivals were associated with higher odds of inter-regional utilization, suggesting more active inter-regional movement in response to capacity or resource limitations.

## Discussion

This nationwide study analyzed patterns of EMS region utilization among critically ill emergency patients in South Korea. More than one in five patients received care outside their residential EMS region, suggesting substantial geographic disparities in access to emergency services. Multivariable analyses indicated that male sex, younger and middle age,

**Table 3. Comparison of general characteristics of critically ill emergency patients by inter-regional utilization status.**

| Variable | Total N (%) | | Inter-regional Utilization | | | | *P*-value |
|---|---|---|---|---|---|---|---|
| | | | Yes | | No | | |
| Total* | 741,701 | (100.0) | 160,700 | (21.7) | 581,001 | (78.3) | |
| Gender | | | | | | | <0.001 |
| Male | 409,537 | (55.2) | 91,759 | (57.1) | 317,778 | (54.7) | |
| Female | 332,164 | (44.8) | 68,941 | (42.9) | 263,223 | (45.3) | |
| Age (yrs) | | | | | | | <0.001 |
| 0–4 | 15,079 | (2.0) | 3,801 | (2.4) | 11,278 | (1.9) | |
| 5–12 | 9,477 | (1.3) | 2,030 | (1.3) | 7,447 | (1.3) | |
| 13–18 | 11,807 | (1.6) | 2,132 | (1.3) | 9,675 | (1.7) | |
| 19–44 | 122,044 | (16.5) | 28,551 | (17.8) | 93,493 | (16.1) | |
| 45–64 | 209,724 | (28.3) | 46,946 | (29.2) | 162,778 | (28.0) | |
| 65–74 | 134,263 | (18.1) | 29,651 | (18.5) | 104,612 | (18.0) | |
| 75–84 | 160,196 | (21.6) | 32,375 | (20.1) | 127,821 | (22.0) | |
| ≥85 | 79,111 | (10.7) | 15,214 | (9.5) | 63,897 | (11.0) | |
| Type of ED | | | | | | | <0.001 |
| REMC | 295,516 | (39.8) | 62,823 | (39.1) | 232,693 | (40.1) | |
| LEMC | 446,185 | (60.2) | 97,877 | (60.9) | 348,308 | (59.9) | |
| Insurance type | | | | | | | <0.001 |
| National health insurance | 621,485 | (83.8) | 137,343 | (85.5) † | 484,142 | (83.3) | |
| Medical aid | 64,273 | (8.7) | 11,058 | (6.9) † | 53,215 | (9.2) | |
| Other | 55,943 | (7.5) | 12,299 | (7.7) † | 43,644 | (7.5) | |
| Illness condition | | | | | | | <0.001 |
| Disease | 534,268 | (72.0) | 118,192 | (73.5) | 416,076 | (71.6) | |
| Non-disease | 207,433 | (28.0) | 42,508 | (26.5) | 164,925 | (28.4) | |
| Route of arrival | | | | | | | <0.001 |
| Direct visit | 538,741 | (72.6) | 103,025 | (64.1) | 435,716 | (75.0) | |
| Transfer from another hospital | 173,504 | (23.4) | 49,908 | (31.1) | 123,596 | (21.3) | |
| Other | 29,456 | (4.0) | 7,767 | (4.8) | 21,689 | (3.7) | |
| Transportation | | | | | | | <0.001 |
| Ambulance | 382,277 | (51.5) † | 83,566 | (52.0) | 298,711 | (51.4) † | |
| Private vehicle | 351,739 | (47.4) † | 75,497 | (47.0) | 276,242 | (47.5) † | |
| Other | 7,685 | (1.0) † | 1,637 | (1.0) | 6,048 | (1.0) † | |
| Triage | | | | | | | <0.001 |
| KTAS 1–2 | 185,737 | (25.0) † | 44,331 | (27.6) | 141,406 | (24.3) † | |
| KTAS 3 | 432,023 | (58.2) † | 92,433 | (57.5) | 339,590 | (58.4) † | |
| KTAS 4–5 | 123,941 | (16.7) † | 23,936 | (14.9) | 100,005 | (17.2) † | |
| Final care outcome | | | | | | | <0.001 |
| Discharge | 579,300 | (78.1) | 122,359 | (76.1) | 456,941 | (78.6) † | |
| Transfer | 99,028 | (13.4) | 24,147 | (15.0) | 74,881 | (12.9) † | |
| Death | 56,637 | (7.6) | 12,797 | (8.0) | 43,840 | (7.5) † | |
| Other | 6,736 | (0.9) | 1,397 | (0.9) | 5,339 | (0.9) † | |
| ED LOS (min) | 267 (144-495) | | 305 (160-581) | | 258 (141-474) | | <0.001 |
| H LOS (day) | 7.2 (3.7-14.8) | | 7.7 (3.8-15.8) | | 7.0(3.7-14.7) | | <0.001 |

Values are presented as patients (%) or median (the first quartile – the third quartile). The * is for row percentage and the others are for column percentage. † The sums of proportions are not equal to 100% due to rounding. ED: Emergency department, REMC: Regional emergency medical center, LEMC: Local emergency medical center, KTAS: Korean triage and acuity scale, LOS: Length of stay, H: Hospital.

**Table 4. Determinants of inter-regional utilization among critically ill emergency patients.**

| Variables | OR (95% CI) | aRR (95% CI) | AME (%p, 95% CI) |
|---|---|---|---|
| Gender | | | |
| Male | 1.08 (1.07, 1.10) | 1.05 (1.03, 1.08) | −8.29 (−12.42, −4.16) |
| Female | Ref | Ref | Ref |
| Age (yrs) | | | |
| 0–4 | 1.16 (1.11, 1.21) | 1.32 (1.04, 1.69) | 1.60 (−2.61, 5.80) |
| 5–12 | 1.06 (1.00, 1.12 | 1.18 (0.87, 1.59) | 2.18 (−1.56, 5.92) |
| 13–18 | 0.90 (0.85, 0.95) | 1.02 (0.80, 1.29) | 1.61 (−1.72, 4.93) |
| 19–44 | 1.35 (1.32, 1.39) | 1.29 (1.15, 1.44) | 3.70 (−0.33, 7.73) |
| 45–64 | 1.24 (1.21, 1.27) | 1.20 (1.07, 1.34) | 1.32 (−1.83, 4.47) |
| 65–74 | 1.13 (1.11, 1.16) | 1.15 (1.00, 1.34) | −0.90 (−3.39, 1.58) |
| 75–84 | 1.02 (1.00, 1.04) | 1.04 (0.96, 1.13) | −5.78 (−7.64, −3.92) |
| ≥85 | Ref | Ref | Ref |
| Type of ED | | | |
| REMC | 1.25 (1.23, 1.26) | 0.90 (0.70, 1.17) | 1.14 (0.67, 1.62) |
| LEMC | Ref | Ref | Ref |
| Insurance type | | | |
| National health insurance | 0.82 (0.80, 0.84) | 0.96 (0.85, 1.08) | 6.14 (−0.71, 12.99) |
| Medical aid | 0.65 (0.63, 0.67) | 0.75 (0.67, 0.83) | 3.34 (−3.66, 10.33) |
| Other | Ref | Ref | Ref |
| Illness condition | | | |
| Disease | 0.90 (0.89, 0.92) | 1.04 (0.89, 1.21) | 1.40 (0.84, 1.95) |
| Non, disease | Ref | Ref | Ref |
| Route of arrival | | | |
| Direct visit | 0.64 (0.62, 0.66) | 0.70 (0.64, 0.76) | 5.39 (2.83, 7.96) |
| Transfer from another hospital | 1.20 (1.16, 1.24) | 1.06 (0.90, 1.24) | 3.74 (0.86, 6.63) |
| Other | Ref | Ref | Ref |
| Transportation | | | |
| Ambulance | 1.19 (1.12, 1.26) | 1.11 (0.91, 1.35) | 2.91 (−0.73, 6.54) |
| Private vehicle | 1.04 (0.98, 1.10) | 1.08 (0.92, 1.26) | 0.78 (−0.87, 2.42) |
| Other | Ref | Ref | Ref |
| Triage | | | |
| KTAS 1, 2 | 1.21 (1.18, 1.23) | 1.19 (0.98, 1.43) | −2.17 (−8.05, 3.72) |
| KTAS 3 | 1.11 (1.09, 1.13) | 1.07 (0.91, 1.25) | 0.71 (−2.71, 4.13) |
| KTAS 4, 5 | Ref | Ref | Ref |
| Final care outcome | | | |
| Discharge | 1.01 (0.95, 1.07) | 1. 01 (0.88, 1.15) | 0.18 (−2.59, 2.95) |
| Transfer | 1.25 (1.17, 1.33) | 1.14 (0.99, 1.32) | 3.03 (−0.48, 6.54) |
| Death | 1.12 (1.05, 1.19) | 1.07 (0.93, 1.22) | 1.36 (−1.78, 4.49) |
| Other | Ref | Ref | Ref |

OR: Odds ratio, CI: Confidence interval, aRR: Adjusted risk ratio, AME: Average marginal effect, ED: Emergency department, REMC: Regional emergency medical center, LEMC: Local emergency medical center, KTAS: Korean triage and acuity scale.

**Table 5. Determinants of inter-regional utilization among critically ill emergency patients: Seoul Southeast, Gwangju, and Daegu EMS regions.**

| Variables | OR (95% CI) | | |
|---|---|---|---|
| | Seoul Southeast | Gwangju | Daegu |
| Gender | | | |
| Male | 1.06 (1.02, 1.10) | 1.12 (1.05, 1.19) | 0.96 (0.91, 1.01) |
| Female | Ref | Ref | Ref |
| Age (yrs) | | | |
| 0–4 | 1.22 (1.10, 1.36) | 1.65 (1.27, 2.15) | 1.75 (1.48, 2.07) |
| 5–12 | 1.13 (0.98, 1.31) | 1.43 (1.08, 1.90) | 1.72 (1.34, 2.19) |
| 13–18 | 1.06 (0.92, 1.22) | 1.18 (0.90, 1.55) | 1.38 (1.09, 1.75) |
| 19–44 | 1.19 (1.11, 1.28) | 1.20 (1.06, 1.37) | 1.88 (1.68, 2.09) |
| 45–64 | 1.41 (1.32, 1.51) | 1.22 (1.09, 1.37) | 1.20 (1.08, 1.33) |
| 64–74 | 1.41 (1.32, 1.51) | 1.17 (1.04, 1.32) | 1.18 (1.06, 1.31) |
| 75–84 | 1.20 (1.12, 1.28) | 1.11 (0.99, 1.25) | 1.15 (1.04, 1.28) |
| ≥85 | Ref | Ref | Ref |
| Type of ED | | | |
| REMC | 1.22 (1.16, 1.29) | 1.70 (1.57, 1.84) | 0.95 (0.90, 1.00) |
| LEMC | Ref | Ref | Ref |
| Insurance type | | | |
| National health insurance | 0.96 (0.88, 1.05) | 1.14 (0.98, 1.32) | 0.90 (0.78, 1.04) |
| Medical aid | 0.84 (0.75, 0.93) | 0.82 (0.69, 0.98) | 0.54 (0.45, 0.64) |
| Other | Ref | Ref | Ref |
| Illness condition | | | |
| Disease | 1.23 (1.17, 1.29) | 0.98 (0.91, 1.07) | 0.92 (0.85, 0.99) |
| Non, disease | Ref | Ref | Ref |
| Route of arrival | | | |
| Direct visit | 0.45 (0.42, 0.49) | 0.41 (0.35, 0.48) | 0.44 (0.40, 0.49) |
| Transfer from another hospital | 0.74 (0.68, 0.80) | 1.51 (1.30, 1.76) | 1.21 (1.09, 1.35) |
| Other | Ref | Ref | Ref |
| Transportation | | | |
| Ambulance | 0.72 (0.59, 0.87) | 1.08 (0.89, 1.32) | 1.36 (1.03, 1.78) |
| Private vehicle | 0.87 (0.72, 1.05) | 1.08 (0.90, 1.31) | 1.50 (1.15, 1.96) |
| Other | Ref | Ref | Ref |
| Triage | | | |
| KTAS 1, 2 | 1.48 (1.39, 1.57) | 1.10 (0.96, 1.25) | 1.07 (0.94, 1.22) |
| KTAS 3 | 1.42 (1.35, 1.50) | 1.15 (1.02, 1.30) | 1.03 (0.92, 1.16) |
| KTAS 4, 5 | Ref | Ref | Ref |
| Final care outcome | | | |
| Discharge | 0.94 (0.73, 1.21) | 1. 26 (0.89, 1.77) | 1. 75 (1.08, 2.82) |
| Transfer | 0.90 (0.69, 1.16) | 1.42 (1.01, 2.01) | 1.68 (1.03, 2.72) |
| Death | 1.01 (0.78, 1.31) | 1.44 (1.00, 2.06) | 1.61 (0.99, 2.62) |
| Other | Ref | Ref | Ref |

OR: Odds ratio, CI: Confidence interval, ED: Emergency department, REMC: Regional emergency medical center, LEMC: Local emergency medical center, KTAS: Korean triage and acuity scale.

ambulance or private vehicle transport, and inter-hospital transfers were associated with higher odds of inter-regional ED utilization. In contrast, Medical Aid patients consistently exhibited lower odds, suggesting potential barriers to access among socioeconomically vulnerable groups. High-acuity patients (KTAS 1–3) and those whose final disposition was transfer also showed greater likelihood of inter-regional care.

Region-specific analyses showed that determinants of inter-regional utilization varied by context. In Seoul Southeast, male, middle-aged, and high-acuity patients were more likely to receive inter-regional care. In Daegu, younger patients and those transferred or arriving by private or ambulance transport had higher odds, whereas in Gwangju, REMC visits and inter-hospital transfers were strongly associated with inter-regional utilization. Pediatric patients also showed elevated odds in Gwangju, reflecting gaps in regional pediatric emergency capacity. Across all three regions, Medical Aid patients consistently had lower odds, which may reflect systemic inequities that persist nationwide. These patterns are in line with previous studies that that uneven resource distribution, institutional capacity, and socioeconomic conditions influence patient flows [14–19].

Multivariable analyses revealed notable differences between relative and absolute effect measures. Male patients showed higher adjusted odds of inter-regional utilization; however, AMEs demonstrated that men had significantly lower absolute predicted probabilities than women. This divergence is consistent with methodological literature indicating that ORs and AMEs may not align when baseline probabilities differ, underscoring the need to present multiple effect measures for accurate interpretation [20–22]. Importantly, presenting both relative and absolute estimates allows policymakers and clinicians to better understand population-level impacts and design EMS interventions that balance equity and efficiency.

Age-related differences also aligned with previous work. Younger and middle-aged adults—particularly those in the 19–44 and 45–64 year groups—showed higher rates of inter-regional utilization, whereas older adults were less likely to receive care outside their EMS region. In our analysis, AMEs for younger groups were positive but not statistically significant, whereas older age groups demonstrated significant decreases in absolute probability, indicating reduced mobility and greater dependence on local services among older adults. Notably, the reduction was statistically significant only in the 75–84-year group, whereas the decrease observed in the 65–74-year group was not significant. These patterns may reflect fewer mobility limitations and greater access to healthcare information among younger adults, and more active healthcare-seeking behavior among middle-aged patients [16]. Pediatric patients also demonstrated elevated utilization, potentially reflecting shortages of pediatric emergency specialists. A recent multilevel analysis reported that transfers of high-acuity pediatric patients were associated not only with severity but also with institutional capacity and geographic distribution of services [17]. Despite initiatives such as the expansion of Dalbit Children's Hospitals, further improvements in accessibility and awareness appear necessary. Similar associations have been reported in surgical care, where age, urbanization, and institutional trust influenced utilization outside residential areas [18].

Insurance status emerged as another important determinant. Patients covered by Medical Aid consistently had lower odds of inter-regional utilization compared with those covered by other insurance types, suggesting that socioeconomic constraints limit access to higher-level emergency facilities. Our AME findings further showed significantly lower absolute probabilities among Medical Aid beneficiaries, reinforcing the interpretation that financial and social vulnerability constrain inter-regional mobility. This pattern underscores how financial and social vulnerability can influence healthcare-seeking behavior and contribute to disparities in emergency care access. Patients with disease-related conditions were more likely to use inter-regional EDs than those with trauma or non-disease-related conditions [19]. This may be related to differences in patient intent, as those with chronic or complex illnesses often seek tertiary hospitals, whereas trauma patients are triaged to designated trauma centers.

System-level factors were also important. Ambulance and private vehicle arrivals were associated with greater odds of inter-regional utilization, although their AMEs were not statistically significant—suggesting that relative differences in transport patterns do not necessarily translate to meaningful absolute increases in utilization. Although inter-hospital

transfers showed higher relative odds, direct visits demonstrated a larger absolute probability of inter-regional use, indicating that patient-driven decisions and system-driven transfers may operate through different pathways. Inter-hospital transfer was a strong predictor as well, which may suggest limitations in the capacity or specialization of initial facilities. Together, these findings highlight inefficiencies in referral and triage systems that warrant further attention at the policy level.

Patients who received inter-regional care had significantly longer ED and hospital lengths of stay compared with those treated within their residential region. Although these associations cannot be interpreted causally, they raise concerns that delays in timely definitive care may adversely affect outcomes. Supporting this, recent Korean studies have shown that EDs implementing structured length-of-stay management systems for critically ill patients achieved lower in-hospital mortality and shorter hospitalization durations [20,23].

Quantitative measures of capacity alone do not appear sufficient to explain these patterns. According to the National Health and Medical Care Statistics Survey, while the total number of hospital beds is concentrated in Seoul and Gyeonggi, the number of beds per 1,000 people is higher in non-metropolitan areas, including Gwangju (21.4), Jeonnam (21.4), Busan (20.5), Jeonbuk (20.5), Gyeongnam (18.9), and Gyeongbuk (16.3). Although hospital beds are concentrated in metropolitan regions, the number of beds per 1,000 residents is higher in many non-metropolitan areas [14]. This suggests that qualitative factors—such as accessibility, specialization, and institutional trust—may influence decisions to seek emergency care outside the residential region. Supporting this, a recent nationwide study found that health service areas without EDs of ≥300 beds had higher 30-day mortality among patients with severe conditions [15].

Although patient preference for large hospitals may partially contribute to inter-regional utilization, regional disparities in emergency care infrastructure—rather than preference alone—appear to be the predominant drivers. Despite government efforts to increase healthcare resources in underserved areas [24,25], substantial inequities persist. To address these challenges, the Ministry of Health and Welfare proposed the 4th National Emergency Medical Plan (2023–2027), which emphasizes a regionally integrated emergency care system based on hospital cooperation. These initiatives aim to improve equity and timely access across EMS regions, though their implementation and effectiveness require ongoing monitoring and evaluation. [26].

This study has several limitations. First, it relied on secondary data from the NEDIS for 2021, which is based on registered residential addresses rather than the specific locations of emergency occurrence. As such, potential misclassifications may have occurred, particularly among students, the elderly, and socially vulnerable populations. Second, the NEDIS database lacks detailed information on inter-hospital transfer processes, such as the reason for transfer, the referring and receiving institutions, and transport times, preventing us from directly assessing the impact on inter-regional ED use. For a more comprehensive analysis, future research should consider integrating NEDIS data with the 119 EMS transport records, National Health Insurance claims, and hospital referral datasets. Third, while this study used regression analysis to identify factors associated with inter-regional utilization, socioeconomic status (e.g., income, education), hospital capacity, or patient preferences, all of which are likely to influence healthcare-seeking behavior, were not included. Future studies should analyze these factors for more nuanced analyses. Finally, this study was based on a single year of cross-sectional data, restricting our ability to evaluate temporal trends or causal relationships. Moreover, subsequent policy initiatives and system reforms (2022–2025), including the 4th National Emergency Medical Plan (2023–2027), may already be affecting inter-regional patient flows, which could not be captured in our analysis.

## Conclusions

This nationwide study found that about one in five critically ill emergency patients in South Korea received care outside their residential EMS region. Inter-regional utilization was more likely among men, younger and middle-aged adults, high-acuity patients, and those transported by ambulance or transferred from another hospital, while Medical Aid patients were less likely to do so. These findings emphasize the need to strengthen local emergency capacity, optimize referral

and transfer pathways, and implement region-specific strategies alongside nationwide reforms to ensure timely and equitable access to emergency services. Future efforts should also focus on the systematic implementation and evaluation of such initiatives to achieve measurable improvements in equity and timely access. Although this study focused on South Korea, the findings have broader implications for other countries facing similar geographic and socioeconomic disparities in emergency care.

## Author contributions

**Data curation:** Mi Ra Oh, Se Hyung Kim.

**Funding acquisition:** Young Jin Huh.

**Methodology:** Young Jin Huh.

**Project administration:** Han Na Lee.

**Resources:** Young Jin Huh.

**Software:** Mi Ra Oh.

**Supervision:** Sung Min Lee.

**Validation:** Han Na Lee.

**Visualization:** Se Hyung Kim.

**Writing – original draft:** Mi Ra Oh, Sung Min Lee.

**Writing – review & editing:** Sung Min Lee.

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
