## [Decision Letter · Decision Letter 0]

4 Sep 2025

PONE-D-25-34614Inter-regional disparities in emergency department utilization among critically ill patients: a nationwide study from South KoreaPLOS ONE

Dear Dr. Lee,

Thank you for submitting your manuscript to PLOS ONE. After careful consideration, we feel that it has merit but does not fully meet PLOS ONE’s publication criteria as it currently stands. Therefore, we invite you to submit a revised version of the manuscript that addresses the points raised during the review process.

We look forward to receiving your revised manuscript.

Kind regards,

Majed Sulaiman Alamri, PhD

Academic Editor

PLOS ONE

**Journal Requirements:**

1. When submitting your revision, we need you to address these additional requirements. Please ensure that your manuscript meets PLOS ONE's style requirements, including those for file naming. The PLOS ONE style templates can be found at https://journals.plos.org/plosone/s/file?id=wjVg/PLOSOne_formatting_sample_main_body.pdf and https://journals.plos.org/plosone/s/file?id=ba62/PLOSOne_formatting_sample_title_authors_affiliations.pdf 2. Thank you for stating in your Funding Statement: This research was supported by a grant of National Medical Center, Republic of Korea (grant number: NMC2023-PR-06).  Please provide an amended statement that declares *all* the funding or sources of support (whether external or internal to your organization) received during this study, as detailed online in our guide for authors at http://journals.plos.org/plosone/s/submit-now. Please also include the statement “There was no additional external funding received for this study.” in your updated Funding Statement. Please include your amended Funding Statement within your cover letter. We will change the online submission form on your behalf. 3. If the reviewer comments include a recommendation to cite specific previously published works, please review and evaluate these publications to determine whether they are relevant and should be cited. There is no requirement to cite these works unless the editor has indicated otherwise. 

Reviewers' comments:

Reviewer's Responses to Questions

**Comments to the Author**

1. Is the manuscript technically sound, and do the data support the conclusions?

Reviewer #1: Yes

Reviewer #2: Yes

2. Has the statistical analysis been performed appropriately and rigorously? 

Reviewer #1: Yes

Reviewer #2: Yes

3. Have the authors made all data underlying the findings in their manuscript fully available?

Reviewer #1: Yes

Reviewer #2: Yes

4. Is the manuscript presented in an intelligible fashion and written in standard English?

Reviewer #1: Yes

Reviewer #2: Yes

5. Review Comments to the Author

**Reviewer #1:** Thank you for the opportunity to review your manuscript which addresses and important and timely issue in healthcare delivery. Below are some suggestions to help improve your work as you ready it for publication.

The paper is well written and flows well.

Introduction: The readers are slowly introduced to the importance of the topic just before the purpose of the study is presented.

Literature Review/Conceptual Framework: There is neither a thorough literature review nor a conceptual framework to justify the hypothesis. I recommend that the authors choose one of the two for a better foundation/justification of their hypothesis.

Also, given that your hypotheses are not even known, it is hard to tell what your Dependent is and what independent and controls are. This could be addressed in a sub-section in your method section

Methods:

It would be helpful to the reader to clarify what you mean by other for route of visit.

While the logistic regression in appropriate for your regression analysis, combining your logistic regression with a marginal effect will enhance your results. With the logistic regression you can only tell whether the odds are higher or lower. With the marginal effect in addition to the odds you will be able to give the percentage by which these odds are higher or lower. This will make it more quantifiable and enhance readability and comprehension for readers. Particularly those that may not be very familiar with your topic and methodology.

It will be important to report the specific statistical analysis you are reporting in your results.

It will be important to explain why the use of both Wilcoxon-Mann-Whitney test and the logistic regression.

Why not contrast the listed regions beds per 1,000 people to that of Seoul and Gyeonggi (line 244-246).

**Reviewer #2:** Overall:

The study uses a large national dataset and shows important disparities. The question is relevant. However, the analysis is limited to one year (2021, COVID period) and policy conclusions must be more cautious.

Major comments

Causality: The cross-sectional design and methods (chi-square, Mann–Whitney U, logistic regression) can only show associations, not causation. Policy recommendations should be phrased more carefully.

Time context: Only 2021 data are analyzed. Please discuss that later policy changes (2022–2025) may already have influenced patient flows.

Age grouping: The 19–64 group is too broad. More detailed age categories or sensitivity analysis would be stronger.

Regression model: Patients are nested in EMS regions, but clustering is not considered. Mixed-effects models or robust SEs should be added. Odds ratios may also exaggerate effects at 22 % prevalence; marginal effects or robust Poisson regression would be helpful.

Model specification: Some variables (triage, final outcome) may be mediators. Please clarify model structure and provide diagnostics (multicollinearity, fit, AUC, calibration).

LOS interpretation: Differences in ED and hospital LOS are descriptive only, not causal. Please present carefully.

Minor comments

Check tables for possible errors (Table 3 numbers, Table 6 CI).

Use consistent terminology (“out-of-region” vs. “inter-regional”).

Add units in tables and STROBE checklist reference.

6. PLOS authors have the option to publish the peer review history of their article (what does this mean?). If published, this will include your full peer review and any attached files.

Reviewer #1: No

Reviewer #2: No

---

## [Author Response · Author response to Decision Letter 1]

24 Sep 2025

We sincerely appreciate the thoughtful and constructive comments provided on our manuscript. We carefully reviewed all suggestions and revised the manuscript accordingly. Below, we provide a detailed response to each comment, with references to the specific changes made in the revised text.

---

## [Decision Letter · Decision Letter 1]

2 Nov 2025

PONE-D-25-34614R1Inter-regional disparities in emergency department utilization among critically ill patients: a nationwide study from South KoreaPLOS ONE

Dear Dr. Lee,

Thank you for submitting your manuscript to PLOS ONE. After careful consideration, we feel that it has merit but does not fully meet PLOS ONE’s publication criteria as it currently stands. Therefore, we invite you to submit a revised version of the manuscript that addresses the points raised during the review process.

Please address reviewer’s comments. Please submit your revised manuscript by Dec 17 2025 11:59PM. If you will need more time than this to complete your revisions, please reply to this message or contact the journal office at plosone@plos.org. Please include the following items when submitting your revised manuscript:

We look forward to receiving your revised manuscript.

Kind regards,

Majed Sulaiman Alamri, PhD

Academic Editor

PLOS ONE

Journal Requirements:

Reviewers' comments:

Reviewer's Responses to Questions

**Comments to the Author**

1. If the authors have adequately addressed your comments raised in a previous round of review and you feel that this manuscript is now acceptable for publication, you may indicate that here to bypass the “Comments to the Author” section, enter your conflict of interest statement in the “Confidential to Editor” section, and submit your "Accept" recommendation.

Reviewer #1: All comments have been addressed

Reviewer #2: (No Response)

2. Is the manuscript technically sound, and do the data support the conclusions?

Reviewer #1: Yes

Reviewer #2: Yes

3. Has the statistical analysis been performed appropriately and rigorously? 

Reviewer #1: Yes

Reviewer #2: N/A

4. Have the authors made all data underlying the findings in their manuscript fully available?

Reviewer #1: No

Reviewer #2: Yes

5. Is the manuscript presented in an intelligible fashion and written in standard English?

Reviewer #1: Yes

Reviewer #2: Yes

6. Review Comments to the Author

Reviewer #1: The authors addressed my concerns and made the paper stronger. The paper brings an important contribution to extant knowledge.

Reviewer #2: The current age categories (0–18, 19–64, ≥65) are not adequate and obscure key heterogeneity. Comparable studies use finer bands (e.g., 0–4, 5–12, 13–18; 19–44, 45–64, 65–74, 75–84, ≥85). Please re-bin and re-run primary analyses (or model age continuously with splines) and update tables/results accordingly.

7. PLOS authors have the option to publish the peer review history of their article (what does this mean?). If published, this will include your full peer review and any attached files.

Reviewer #1: No

Reviewer #2: No

---

## [Author Response · Author response to Decision Letter 2]

7 Nov 2025

We sincerely appreciate the thoughtful and constructive comments provided on our manuscript. We carefully reviewed all suggestions and revised the manuscript accordingly. Below, we provide a detailed response to each comment, with references to the specific changes made in the revised text.

Reviewer #1:

1. The authors addressed my concerns and made the paper stronger. The paper brings an important contribution to extant knowledge.

Answer: Thank you for your thoughtful review.

Reviewer #2:

1. The current age categories (0–18, 19–64, ≥65) are not adequate and obscure key heterogeneity. Comparable studies use finer bands (e.g., 0–4, 5–12, 13–18; 19–44, 45–64, 65–74, 75–84, ≥85). Please re-bin and re-run primary analyses (or model age continuously with splines) and update tables/results accordingly.

Answer:

Thank you for this important suggestion. We fully agree that age effects may be heterogeneous, particularly among pediatric and older adult subgroups.

In response, we re-classified age into eight standard epidemiologic groups

(0–4, 5–12, 13–18, 19–44, 45–64, 65–74, 75–84, and ≥85 years),

and re-estimated all primary models (OR, aRR, and AME).

This refinement revealed more granular and clinically coherent patterns of inter-regional utilization. As reflected in the updated Tables 3–5:

• Younger patients (0–4 and 5–12 years) and

middle-aged adults (19–44 and 45–64 years) had higher odds of inter-regional utilization.

• Older adults (≥75 years)—particularly those ≥85 years—were least likely to receive care outside their EMS region.

• Re-estimation of all models produced modest shifts in effect sizes, largely due to redistribution across narrowed age bands.

We revised the Abstracts, Methods, Results, and Discussion to reflect the updated classification and interpretation.

---

## [Decision Letter · Decision Letter 2]

25 Nov 2025

Inter-regional disparities in emergency department utilization among critically ill patients: a nationwide study from South Korea

PONE-D-25-34614R2

Dear Dr. Lee,

We’re pleased to inform you that your manuscript has been judged scientifically suitable for publication and will be formally accepted for publication once it meets all outstanding technical requirements.

Kind regards,

Majed Sulaiman Alamri, PhD

Academic Editor

PLOS ONE

Additional Editor Comments (optional):

Reviewers' comments:

Reviewer's Responses to Questions

**Comments to the Author**

1. If the authors have adequately addressed your comments raised in a previous round of review and you feel that this manuscript is now acceptable for publication, you may indicate that here to bypass the “Comments to the Author” section, enter your conflict of interest statement in the “Confidential to Editor” section, and submit your "Accept" recommendation.

Reviewer #1: All comments have been addressed

Reviewer #2: All comments have been addressed

2. Is the manuscript technically sound, and do the data support the conclusions?

Reviewer #1: Yes

Reviewer #2: Yes

3. Has the statistical analysis been performed appropriately and rigorously? 

Reviewer #1: Yes

Reviewer #2: Yes

4. Have the authors made all data underlying the findings in their manuscript fully available?

Reviewer #1: No

Reviewer #2: Yes

5. Is the manuscript presented in an intelligible fashion and written in standard English?

Reviewer #1: Yes

Reviewer #2: Yes

6. Review Comments to the Author

Reviewer #1: The authors have adequately addressed my concerns. I do not have any additional concerns to be addressed at this time. Thank you for your diligence.

Reviewer #2: The revision has been addressed well and comprehensively. The authors have adequately responded to previously raised points and have substantially improved the overall quality of the manuscript. Conceptual ambiguities have been resolved, methodological aspects clarified, and the presentation has been strengthened.

I have no further comments or concerns regarding research ethics, publication ethics, or potential dual publication. In my view, the manuscript is now suitable for publication.

7. PLOS authors have the option to publish the peer review history of their article (what does this mean?). If published, this will include your full peer review and any attached files.

Reviewer #1: No

Reviewer #2: No

---

## [Editor Report · Acceptance letter]

PONE-D-25-34614R2

PLOS ONE

Dear Dr. Lee,

I'm pleased to inform you that your manuscript has been deemed suitable for publication in PLOS ONE. Congratulations! Your manuscript is now being handed over to our production team.

Kind regards,

on behalf of

Prof. Majed Sulaiman Alamri

Academic Editor

PLOS ONE